# Feasibility Study of Combining Hyperspectral Imaging with Deep Learning for Chestnut-Quality Detection

**DOI:** 10.3390/foods12102089

**Published:** 2023-05-22

**Authors:** Qiongda Zhong, Hu Zhang, Shuqi Tang, Peng Li, Caixia Lin, Ling Zhang, Nan Zhong

**Affiliations:** 1College of Engineering, South China Agricultural University, Guangzhou 510642, China; 20223171034@stu.scau.edu.cn (Q.Z.);; 2Heyuan Branch, Guangdong Laboratory for Lingnan Modern Agriculture, Heyuan 517000, China; 3Guangdong Provincial Key Laboratory of Agricultural Artificial Intelligence (GDKL-AAI), Guangzhou 510642, China; 4College of Biology and Food Engineering, Guangdong University of Petrochemical Technology, Maoming 525000, China

**Keywords:** chestnut, hyperspectral imaging, quality detection, deep learning, important wavelengths

## Abstract

The rapid detection of chestnut quality is a critical aspect of chestnut processing. However, traditional imaging methods pose a challenge for chestnut-quality detection due to the absence of visible epidermis symptoms. This study aims to develop a quick and efficient detection method using hyperspectral imaging (HSI, 935–1720 nm) and deep learning modeling for qualitative and quantitative identification of chestnut quality. Firstly, we used principal component analysis (PCA) to visualize the qualitative analysis of chestnut quality, followed by the application of three pre-processing methods to the spectra. To compare the accuracy of different models for chestnut-quality detection, traditional machine learning models and deep learning models were constructed. Results showed that deep learning models were more accurate, with FD-LSTM achieving the highest accuracy of 99.72%. Moreover, the study identified important wavelengths for chestnut-quality detection at around 1000, 1400 and 1600 nm, to improve the efficiency of the model. The FD-UVE-CNN model achieved the highest accuracy of 97.33% after incorporating the important wavelength identification process. By using the important wavelengths as input for the deep learning network model, recognition time decreased on average by 39 s. After a comprehensive analysis, FD-UVE-CNN was deter-mined to be the most effective model for chestnut-quality detection. This study suggests that deep learning combined with HSI has potential for chestnut-quality detection, and the results are encouraging.

## 1. Introduction

Chestnuts have a long history as an agricultural product and are widely distributed worldwide, but mainly concentrated in Asia and Europe. China is a major producer and consumer of chestnuts, with production reaching 295,661 ha in 2021, according to FAO. Chestnuts are rich in nutritional value, consisting of 42.2–66.5% starch, 40.3–60.1% moisture, 9.5–23.4% total sugar, 4.8–9.6% crude protein, 2.2–3.7% crude fiber, 1.8–3.4% ash, 2.8–3.2% fat, and essential amino acids and minerals [1]. In recent years, the demand for chestnuts has significantly increased due to the growing popularity of cooked chestnuts and various chestnut-based foods. However, there are numerous low-quality chestnuts in the market that are affected by factors such as planting terrain, ripening time, and storage conditions. These low-quality chestnuts can pose a serious health risk to consumers after processing and reaching the market.

In southern China, most chestnuts are planted in the hills and mountains, and these chestnuts are planted by farmers and purchased by processing companies from the farmers without forming large-scale plantation industrial parks, and the ripening time of chestnuts varies. Therefore, due to the lack of large-scale planting and inconsistent ripening time of chestnuts, the quality of chestnuts procured by processing enterprises varies. As a result, processing enterprises have an urgent need for technology that can identify the quality of chestnuts.

The quality of chestnuts is affected by factors such as mold, stiffness, sweetness, insects, and many others. Among these factors, mold has the greatest impact on the chestnut. The moisture content of chestnuts after harvest will change with storage time and environment, and they are susceptible to mold and mildew during storage and transportation due to their rich nutrition and moisture [2]. Compared to other agricultural products, because the chestnut has a hard shell and chestnut mold often occurs inside the chestnut, there is no obvious change in the appearance; it is difficult to distinguish the internal mold of the chestnut by the naked eye [3], which brings great challenges to the processing and eating quality of chestnut.

Typically, several techniques are now available to determine the quality of chestnuts, including machine vision [4], near-infrared spectroscopy (NIR) [1], and computed tomography (CT) [3]. Machine vision technology is widely used to detect damage on the surface of agricultural products, but damage to chestnuts often occurs internally, and cannot be detected by machine vision technology when damage occurs inside the chestnut. Good results were obtained using X-ray–CT chestnut defect experiments [3,5]. For example, the accuracy of identifying healthy, severely, and slightly defective chestnuts was 0.929, 0.937, and 0.836, respectively [5]. However, it cannot yet be applied to large-scale chestnut identification and production classification because CT requires theoretical analysis by professionals, which is costly and inefficient. In addition, HIS has a unique advantage over X-ray–CT in that it can represent changes in reflectance in the near-infrared range as data in the form of images [6]. The optical properties of chestnuts vary from quality to quality, and even small changes in the diffuse reflectance coefficient due to optical properties can be detected directly. This technique is more sensitive to slight changes in chestnut quality than X-ray–CT using HIS. NIR has been combined with the chemometric approach for chestnut quality classification [1], where the highest accuracy of 0.96 was achieved using the linear discriminant analysis (LDA) model, but NIR combined with the chemometric approach was not as efficient as the HSI technique for detection.

Differently from the above methods, HSI obtains both spectral and spatial information about the sample over a wide spectral range, while also providing imaging data [7]. It is a non-contact, nondestructive, and rapid detection technique that takes advantage of the fact that different constituents of the sample have different spectral absorption, and the image will reflect significantly on defects at specific wavelengths [8]. In addition, HSI has the advantage of capturing many narrower spectral bands in a continuous spectral range [9]. In recent years, with the rapid development of HSI, many excellent machine learning and deep learning algorithms have been proposed to solve the classification problem of nondestructive inspection of agricultural products. In quality detection [10], the prediction coefficient of determination R^2^ reached 0.84 when using PLSR combined with variable selection for skin egg-quality detection. For lychee-browning detection [11], the highest coefficient of determination was 0.946 when collecting six days of spectral data separately and then establishing six lychee-identification models. In the wheat mildew classification study [12], hyperspectral data were collected from wheat on the second and fifth day, respectively, and an optimal classification accuracy of 0.91 was achieved. Many studies also exist in spectroscopy combined with deep learning such as rice variety identification [13], salmon adulteration identification [7], and heavy metal detection in oilseed rape [14]. These HSI, combined with deep learning models, all have an accuracy of 0.9 or higher, so the application of deep learning algorithm models to identify chestnut quality is a very promising direction.

The overall goal of this study is to explore the feasibility of hyperspectral methods combined with deep learning networks for chestnut-quality classification and to attempt to find an optimal recognition model that can be easily applied to practical processing and production. The main undertakings of this paper are as follows:The feasibility of the hyperspectral method combined with the deep learning model for chestnut quality classification was verified.To determine the optimal identification band for different-quality chestnuts, the important bands of chestnuts of varying quality were extracted and compared with the full-band chestnuts. This allowed for verification of the impact of important band classification on chestnut quality.The visualization of chestnut-quality detection by hyperspectral data combined with principal component analysis is presented.Hyperspectral detection has the potential to determine the quality of agricultural products such as grapefruit, lychee, peanut, and mangosteen, which may not be easily observable by the naked eye.

## 2. Materials and Methods

### 2.1. Material Acquisition and Processing

To ensure that the chestnut samples had a uniform level of freshness, all chestnut samples in this study were provided with 170 freshly picked chestnuts by chestnut-processing companies. Chestnuts were transported by courier, and the first day of chestnut storage (28 December 2022) was used as the starting time for chestnut storage, and the number of days was recorded as “1st d”. Chestnut data was obtained by hyperspectral instrumentation on the first day of sample acquisition, and the chestnuts were stored normally after the first phase of chestnut data collection. Chestnuts were stored in semi-enclosed boxes at a temperature and humidity of 13 °C and 67%, respectively, for the first thirty days of storage, and at a temperature and humidity of 20 °C and 80%, respectively, for the second thirty days of storage. The average temperature and humidity were significantly higher in the second thirty days of storage, and the trend of chestnut quality changes would be accelerated when the chestnut was in a higher temperature and humidity environment in the second thirty-day stage [15]. Spectral data of chestnuts were obtained on the 1st day, 30th day, and 60th day of storage, and recorded as fresh, sub-fresh (slightly moldy), and rotten (severely moldy). This data was used for subsequent experimental analysis.

### 2.2. Spectral Image Acquisition and Correction

The HSI system mainly consists of a hyperspectral imaging lens (Specim FX17, Spectral Imaging Ltd., Oulu, Finland), a light source, a mobile platform, and a computer, with the hyperspectral imaging lens operating in the near-infrared band (935–1720 nm). The hyperspectral imaging lens has an image resolution of 640 × 640 px and a spectral resolution of 8 nm, and the light source is a 280 W halogen lamp. In order to obtain clear image information, the height between the lens and the sample is 32 cm, and the specific workflow is shown in Figure 1. During the spectral instrument data acquisition process, there is dark current noise and the effect of uneven illumination [6]. To increase the accuracy of the data, corrections are made by transforming the original image (*I_raw_*) into a reflectance image (*I_c_*) using standard white reference images (*I_white_*) and dark reference images (*I_dark_*):(1)$$I_{c}=\frac{I_{raw}-I_{white}}{I_{white}-I_{raw}}$$
where *I_raw_* is the original hyperspectral image, *I_c_* is the corrected image, *I_white_* is obtained by using a white PTFE rod with almost 100% reflectivity, and *I_dark_* is obtained by completely covering the lens with an opaque cap.

### 2.3. Moisture Loss Measurement

Immediately after each phase of spectral data collection, the samples were weighed using an electronic scale (longbei, Home Electronic Scale, Guangzhou, China), and considering the consistency of each weighed sample, each chestnut sample was labeled with a serial number to ensure that the same sample was measured each time. After each measurement, the sample was stored at room temperature, and the weight of the 1st d sample was measured as $W_{f}$, the weight of 30th and 60th d was measured as *W_d_*, and the formula of moisture loss was calculated as in Equation (2) [16]:(2)$$W_{l}=\frac{W_{f}-W_{d}}{W_{f}}$$

The mean, maximum, and variance data of the experimental sample weights are shown in Table 1.

### 2.4. Principal Component Analysis Method

Principal component analysis (PCA) is an unsupervised data dimensionality-reduction method targeted at exploring the sample space and interpreting the sample space as an effective statistical tool. An excessive number of variables can be transformed into a smaller number of new potential variables, which are also called principal components (PCs) [17]. Each PC has a corresponding score, which is provided in the form of scatter plots which illustrate the association between the samples. These score plots ensure that the spectral data can be interpreted successfully. In this study, the cluster analysis of the spectral data of good-quality chestnuts and poor-quality chestnuts was performed using principal component analysis to achieve a qualitative analysis of chestnut-quality detection.

### 2.5. Spectral Image Pre-Processing

Proper pre-processing of the spectral data can increase the correlated chemical peaks in the spectra and reduce the effects of baseline shift and overall curvature [18]. In this paper, four pre-processing methods are used, including standard normal-variables transformation (SNV), multiplicative scatter correction (MSC), and first-order derivative (FD). The standard normal variables transformation (SNV) can effectively eliminate the scattering problem and provide high-quality data for building the identification model [19]. MSC can effectively eliminate the spectral differences caused by different scattering levels and thus enhance the correlation between the spectra and the data [20]. FD is used to amplify the trend of the spectral images through the derivative processing of the spectral images [21]. In this study, several pre-processing methods are compared to find an optimal pre-processing method.

### 2.6. Feature Selection Algorithm

Hyperspectral data are characterized by a strong correlation between adjacent bands and high redundancy [22]. Appropriate use of feature-extraction methods can effectively reduce the dimensionality of the spectral data and simplify the model. In this study, three feature-extraction algorithms were used: competitive adaptive reweighted sampling (CARS), the successive projections algorithm (SPA), and the uninformative variable elimination (UVE). SPA is a forward-selection algorithm designed for selecting spectral features. It selects the least redundant band from the original spectrum in order to reduce the effect of spectral covariance [23]. CARS is a feature-selection method that combines Moncatello sampling (MC) and partial least-squares (PLS) model regression, and this approach uses cross-validation (CV) for determining the subset with the lowest root-mean-square error [24]. The equation for root-mean-square error cross validation (*RMSECV*) is as follows:(3)$$RMSECV=\sqrt{\frac{1}{n}\sum_{i=1}^{n}{\left(y-y_{c}\right)}^{2}}$$
where *y* denotes the true value and *y_cv_* denotes the predicted value in cv. UVE is a variable selection method based on stability analysis of PLS regression models for eliminating redundant or uninformative spectral parameters [25].

### 2.7. Traditional Machine Learning Methods

The partial least-square discriminant analysis (PLS-DA) model is useful for solving classification problems in two stages. The first stage is the application of PLS components for dimensionality reduction, and the second stage is the predictive model building, i.e., discriminant analysis. In classification, PLS-DA transforms categorical variables into continuous variables, and then calculates the Latent Variable Scores (LVS) to fit the model using covariance [26].

Support vector machine (SVM) is a well-established classification method that mainly solves nonlinear classification, function-estimation, and pattern-recognition problems. The basic idea is to transform the low infinitesimal indistinguishable variables into a hyperplane that can correctly partition the training data set in a high-dimensional feature space with maximum measurement intervals. In this study, the radial basis function (RBF) is chosen as the kernel function, which has a better ability to handle nonlinear data [27]. Random forest (RF) is an effective method for analyzing high-dimensional data by constructing multiple decision trees in parallel to combine their results to produce an output [28], and the decision trees and minimum number of leaves in this paper are set to 2.

### 2.8. Deep Learning Models

#### 2.8.1. Convolutional Neural Networks

CNN is an unsupervised network, which is commonly used in visual image and speech processing, for example. With the continuous research and use of CNN, many applications using CNN for hyperspectral image classification have also emerged in the field of spectroscopy [29]. CNN consists of an input layer, convolutional layer, maximum pooling layer, fully connected layer, and output layer. In this study, a chestnut quality classification model called “1DCNN” is proposed and found to be effective. The structure diagram of the one-dimensional CNN is shown in Figure 2. The 1DCNN model comprises of three convolutional layers with a kernel size of 2 * 1. The number of kernels used were 16, 32, and 32 respectively. The step size and padding were set at 2 and 0, respectively. The maximum pooling layer has a size of 2 * 1 with a step size of 1. The fully connected layer reduces the neuron parameters to 3, and finally the output is passed through softmax. To speed up the training process and avoid the vanishing gradient problem, the rectified linear unit (ReLU) is used as the activation function [30] (Equation (4)). For classification problems, the cross-entropy function (Equation (5)) is often applied as a loss function, and the cross-entropy function to measure the distance between the actual output and the desired output [31].
(4)$$f\left(x\right)=\max\left(0,x\right)$$
(5)$$L_{C}\left(y,\tilde{y}\right)=\frac{1}{N}\overset{S}{\underset{i=1}{\sum }}\left(-\overset{q}{\underset{j=1}{\sum }}y_{j}^{i}\log\left({\tilde{y}}_{j}^{i}\right)\right)$$
where *L_C_* represents the classification loss function; *y* is the label vector; *ỹ* is the predicted vector; *s* is the number of samples; and *q* is the number of classes. In a classification task, the softmax operator is applied to the predicted output to obtain the predicted probability, and the cross-entropy loss is then compared with the true label, which is presented in Equation (6).
(6)$$L=-\underset{i}{\sum }\underset{j}{\sum }y_{i}logp_{ij}p=softmax\left(\tilde{y}\right)$$

#### 2.8.2. Long Short-Term Memory Network

The long short-term memory network (LSTM) is a special kind of recurrent neural network (RNN) that solves the general recurrent neural network long-term dependency problem [32]. LSTM can perform a single operation across all sequence lengths and optimize vanishing gradient problems through its gating features [33]. T Due to the advantages of the LSTM model, it has also been more widely used in the field of spectroscopy in recent years [9,34]. The LSTM model in this study is shown in Figure 2. The LSTM model consists of an input layer, two LSTM layers, and one fully connected layer. Among them, ReLU is chosen for the activation function, thus speeding up the learning speed and avoiding the problem of vanishing gradients. The deep learning model in this study is executed in MATLAB R2022a (Mathworks, Natick, MA, USA).

## 3. Results and Discussion

### 3.1. Change in Water Content

The study found that the moisture content of chestnuts decreased as the number of storage days increased, and the rate of moisture loss was significantly faster during the first 30 days than the last 30 days. Measurements taken of the weight of fresh, sub-fresh, and rotten chestnuts led to this conclusion. The weight change and water loss of chestnuts increased with the increase of storage days, as shown in Figure 3, and water loss slows down for 30–60 days, probably because the chestnut mold develops during this period and mold multiplies rapidly [35]. The average weight of chestnuts decreased by 2.68 g after 60 days of storage at room temperature, and the average moisture loss was 29.1%, which also indicated that there was a large difference in the moisture of rotten chestnuts stored for a long time compared to fresh chestnuts.

### 3.2. Spectral Overview

The average spectra of the different chestnut qualities are presented in Figure 4, with the spectral wavelength range in the NIR band (935–1720 nm). In general, the three different qualities of chestnuts have the same spectral curve waveforms with similar peaks and troughs, but different reflectance values due to the consistent internal composition of the chestnuts [22]. Among them, the sub-fresh and rotten chestnuts showed a more scattered spectral curve, which may be due to the different degrees of quality variation in these chestnuts.

Compared to fresh chestnuts, sub-fresh chestnuts showed a higher reflectance due to the water-absorption peak associated with the O–H bond at 970 and 1450 nm (Figure 4) [36]. The measurement of chestnut moisture loss $W_{l}\;$ also showed that chestnut moisture loss was rapid during the first thirty days of storage, showing a large difference in spectral water absorption peaks. In the second thirty days, it is likely that the moisture was produced by the rapid multiplication of chestnut mold bacteria, as has been seen in some studies on chestnut mold [35]. The appearance of an absorption peak at 1200 may be related to C–H bonded sugars and starch [37]. Although the average spectra of different quality chestnuts exhibit differences, it is clear from the overall spectral map that this is due to large variations in individual chestnuts weighted to the average spectrum. Therefore, a more reliable way to identify chestnuts with different degrees of variation needs to be established.

### 3.3. PCA Qualitative Analysis

Before building the model, the raw spectral data were visualized and qualitatively clustered using PCA. Since the cumulative contribution of the first two principal components (PC1, PC2) reached 97.4%, with PC1 explaining 79.8% of the variance and PC2 explaining 17.6% of the variance, PC1 and PC2 were chosen to explain the clustering analysis of the chestnut spectral data. As shown in the scatter plot of PCA scores in Figure 5, the distribution of the three chestnut species can be effectively separated using confidence ellipses. There is a certain degree of overlap between fresh chestnuts and sub-fresh chestnuts, and a small overlap between sub-fresh chestnuts and rotten chestnuts, which may be caused by the different change rates of different chestnuts.

In the visualization of hyperspectral data, the first three principal component images were obtained by PCA dimensionality reduction; as shown in Figure 6, there was no obvious difference between the original images of fresh and sub-fresh chestnuts, while there was a slight difference in the principal component images, and at this time when moldy chestnuts appeared on the principal component images of sub-fresh chestnuts could be distinguished more intuitively by means of images. When comparing the principal component images of the sub-fresh and moldy chestnuts, there is a significant difference, which further illustrates the feasibility of spectra for chestnut-quality differentiation.

### 3.4. Chestnut Quality Identification Results Using Machine Learning Models

For the three pre-processing methods of RAW, MSC, SNV, and FD spectra (Figure 7), a chestnut-quality classification model was developed at 224 wavelengths. The identification models used PLS-DA, SVM, and RF for fresh (1st d), sub-fresh (30th d), and rotten chestnuts (60th d), respectively. The data are divided into training and test sets in the ratio of 3:1, where the training and test sets for RF are used in a disordered order to improve their generalization ability. The recognition results of PLS-DA, SVM, and RF models for different pre-processing methods are shown in Table 2, and Figure 8 shows the corresponding confusion matrix for the test set with higher accuracy.

**Figure 7 foods-12-02089-f007:**
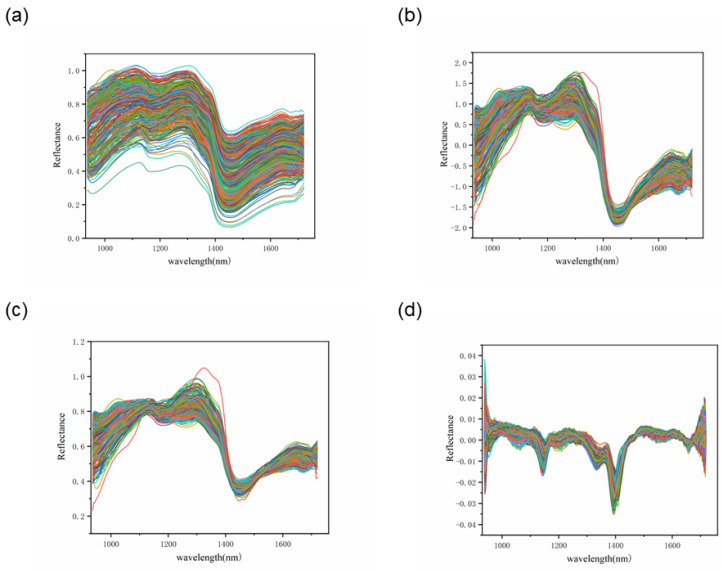
Pre-processing curves: (**a**) RAW; (**b**) SNV; (**c**) MSC; (**d**) FD.

**Table 2 foods-12-02089-t002:** Recognition results of machine learning models based on different pre-processing methods.

Model	Pre-Processing Method							
	RAW		SNV		MSC		FD	
	Train ^2^	Test ^3^	Train	Test	Trian	Test	Trian	Test
PLS-DA	88.61	90.67	88.89	90	88.89	90	93.33	93.33
RF	95.27	88	95.28	89.33	95.56	91.33	96.39	94
SVM	87.45	87.06	93.73	93.33	77.84	76.27	88.43	88.43

^2^ Train set; ^3^ test set.

**Figure 8 foods-12-02089-f008:**
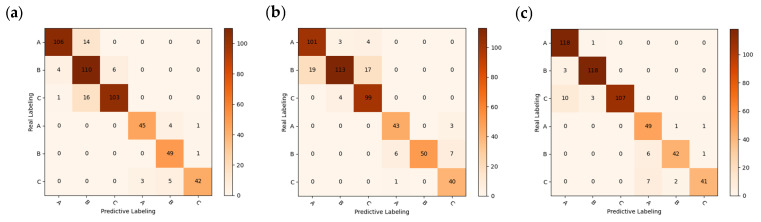
Confusion matrix for different recognition models: (**a**) PLS-DA + RAW; (**b**) SVM + RAW; (**c**) RF + RAW; (**d**) RF + MSC; (**e**) RF + FD; (**f**) SVM + SNV.

It can be observed that in most cases, the use of pre-processing can improve the accuracy of the model for the spectra. The RF + SNV and RF + MSC approaches achieved 95.28% and 95.56% in the training set, respectively, while the RF + FD and SVM + SNV approaches achieved the highest accuracy of 94% and 93.33% in the test set, respectively, with good results in overall accuracy using both PLS-DA and RF models.

The confusion matrix was used to further analyze the degree of misclassification within each model, with A, B, and C labels representing fresh, sub-fresh, and rotten chestnuts, respectively. From an overall perspective, A and B are susceptible to misclassification, while C is also somewhat susceptible to misclassification with B and C. This phenomenon is analogous to the behavior of spectral curves, where fresh and sub-fresh chestnuts exhibit similar curves, but curves of rotten chestnuts are considerably more dispersed, leading to a higher likelihood of misclassification as fresh or sub-fresh chestnuts. It is because of the similarity in spectral curve that it is more difficult for the traditional classification model to obtain satisfactory accuracy directly [9], so further deep learning approaches are needed to obtain satisfactory accuracy.

### 3.5. Chestnut Quality Recognition Results Using Deep Learning Models

The dataset was partitioned using the same method as before, and data was randomized to enhance the model’s generalization ability. The CNN and LSTM deep learning models were built to quickly detect different quality chestnuts, respectively, and the models were trained for 500 rounds using segmented learning with a learning rate of 0.01 for the first 80% iterations and a decreasing learning rate of 0.1 for the last 20%, respectively. The recognition performance of the models is shown in Table 3. Observing the results in the table shows that the use of deep learning models is significantly better than traditional machine learning models, which is not an exception: in potato-disease detection [38], CNN models outperform traditional models in terms of recognition results. The same phenomenon is demonstrated in this experiment, where the CNN model also outperforms the LSTM model in terms of deep learning in general. The CNN models achieved 100% accuracy in the training set after pre-processing, and the highest accuracy of 98.67% was achieved in the MSC-CNN test set. However, the FD-LSTM test set has the highest accuracy of 99.72%, and the model accuracy and loss function are shown in Figure 9.

**Table 3 foods-12-02089-t003:** The results of chestnut-quality recognition based on deep learning models.

Model	Time ^4^	Pre-Processing Method							
		RAW		SNV		MSC		FD	
		Train	Test	Train	Test	Train	Test	Train	Test
CNN	50 s	99.17	96	100	96.67	100	98.67	100	98.33
LSTM	56 s	98.33	95.33	99.44	97.33	98.33	94	99.33	99.72

^4^ Average model running time.

**Figure 9 foods-12-02089-f009:**
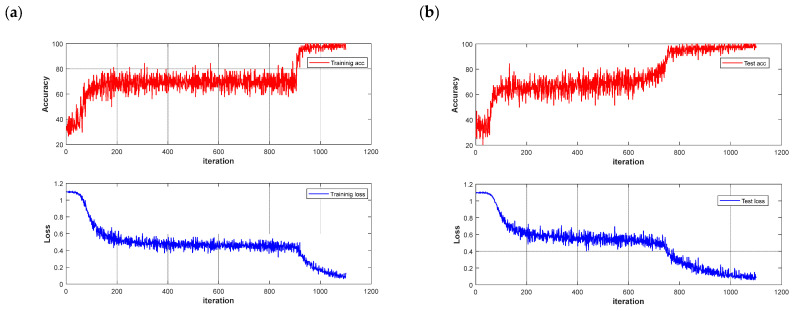
Loss function and accuracy curve of LSTM models: (**a**) training set; (**b**) test set.

Although the recognition accuracy is improved by using a deep learning network, the deep learning network requires constant iterations, and the network has a low recognition efficiency with an average running time of 53 s. Therefore, it is important for practical use to reduce the amount of input information and improve the model recognition efficiency by screening spectral feature wavelengths.

### 3.6. Identification of Important Wavelengths

The accuracy of the traditional classification model does not achieve more satisfactory results, and the deep learning model has good results in terms of recognition accuracy, but the recognition efficiency is low. In this paper, we try to identify important wavelengths by spectral feature selection to improve the recognition efficiency of the deep learning model and improve the accuracy of the traditional model at the same time. The feature selection is performed by the SPA, CARS, and UVE algorithms, and the important wavelengths are recognized by the FD pre-processing method, which performs better in traditional classification models and deep learning models, while RAW is used as a control.

Different algorithms for the extraction results are shown in Figure 10. From the figure, it can be observed that although different feature wavelength-identification algorithms are used, the identification results are mostly concentrated around 1000, 1400, and 1600 nm wavelengths, which can be used as important wavelengths for chestnut = quality identification.

### 3.7. Important Wavelength-Identification Results

RF and CNN, which have higher accuracy in machine learning and deep learning models, were applied in further studies, and the specific results are shown in Table 4 and Table 5. The model confusion matrix is shown in Figure 11, where there is a small improvement in the accuracy of the test set before the identification of important wavelengths using the RF model. The RAW-UVE-CNN recognition model has a slight increase in the accuracy of the test set, and a slight decrease in the accuracy of the test set of FD pre-processing, which is likely to have been caused by the reduction of the network input parameters.

**Table 4 foods-12-02089-t004:** Results of different important wavelength recognition algorithms based on RF.

Method	Variables ^5^	Preprocessing Method			
		RAW		FD	
		Train	Test	Train	Test
SPA	9, 9 ^6^	96.67	91.33	94.44	87.33
CARS	19, 44	93.89	90	96.94	92
UVE	121, 54	97.5	89.66	96.94	90

^5^ The number of input variables in RF model. ^6^ Denotes the RAW spectra and FD pre-processing mode input variables, respectively.

**Table 5 foods-12-02089-t005:** Results of different important wavelength recognition algorithms based on CNN.

Method	Variables	Time	Preprocessing Method			
			RAW		FD	
			Trian	Test	Trian	Test
SPA	9, 9	8 s	98.89	95.33	100	97.33
CARS	19, 44	10 s	98.33	94	99.17	94
UVE	121, 54	11 s	98.61	97.33	99.44	98

**Figure 11 foods-12-02089-f011:**
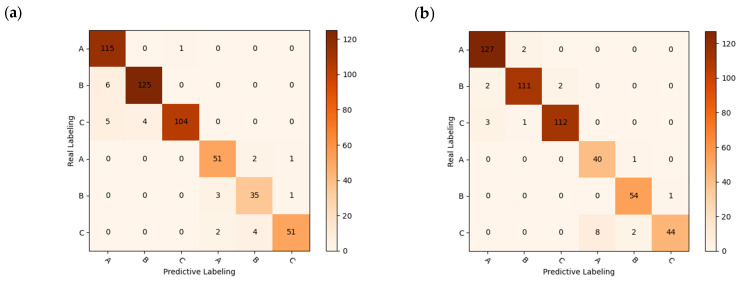
Confusion matrix for RF models: (**a**) RAW-SPA; (**b**) FD-CARS.

However, although there is a slight decrease in accuracy by identifying significant wavelengths using the deep learning network, this approach significantly reduces the recognition time of the network and improves the network recognition efficiency. The model results and confusion matrix are shown in Figure 12. The average running time of the deep learning network was reduced from 50 s to 11 s. The improvement in recognition efficiency offers potential for the practical applications of chestnut-quality detection.

## 4. Discussion

In food processing, it is crucial to identify the quality of chestnuts quickly and non-destructively. There have been previous studies on chestnuts investigating the identification of quality, mold, and origin, and this paper builds on that to conduct a more in-depth study. From the perspective of similar studies, researchers have studied chestnut origin and mold; for example, one investigated the application of backpropagation neural network (BPNN) to achieve the classification of healthy and moldy chestnuts with an accuracy of 0.99 [35]. Chestnut origin identification experiments using 1D-CNN and PLS-DA algorithms both achieved an accuracy of 0.971 [39]. In addition, the overall accuracy of HIS combined with deep learning for classification in nut-quality assessment was 0.958 [40]. This paper offers significant improvements in chestnut quality detection, distinguishing itself from prior research in the following ways. Firstly, a more accurate quality classification is utilized to categorize chestnuts into three groups for detection, with the application of the latest deep learning methodologies that further improve accuracy. Additionally, this paper shows that the moisture loss value of chestnuts is strongly correlated with the 1450 nm water absorption peak that increases as chestnuts are stored over time. Wavelength identification algorithms such as SPA, CARS, and UVE are employed to identify that the key wavelength range for chestnuts is mainly concentrated around 1200, 1400, and 1600 nm. Utilizing the identified wavelength distribution characteristics, multispectral equipment can be developed, making chestnut quality inspection more accessible and efficient.

## 5. Conclusions

This paper utilized a combination of HSI and traditional classification models, as well as deep learning models, to detect chestnut quality. By comparing the recognition effects of different models, combined with different treatments, the best model for chestnut-quality detection and the important wavelengths for chestnut-quality detection were identified. It was observed that FD-RF achieved the highest accuracy of 94% in the traditional classification model, while FD-LSTM achieved the highest accuracy of 99.72% in the deep learning network, with various pre-processing methods and different classification models employed.

Furthermore, feature extraction was used to identify the important wavelengths for chestnut-quality identification, which were mainly distributed around 1000, 1400, and 1600 nm. The significant wavelengths were identified with a slight increase in the original spectral accuracy in the RF model, and the significant wavelengths were used as input for the deep learning network, which had a slight decrease in accuracy but significantly reduced the network identification time (39 s on average).

Overall, the aim of this paper is to investigate the possibility of using hyperspectral imaging in conjunction with a deep learning model for chestnut-quality detection. Through the identification of significant wavelengths, we were able to enhance the effectiveness of the hyperspectral detection method. Furthermore, a FD-UVE-CNN model was developed, which showed high levels of accuracy and recognition efficiency with percentages of 97.33% and 11 s, respectively. These findings suggest that deep learning models have the potential to be applied in the practical detection of chestnut quality.

## Figures and Tables

**Figure 1 foods-12-02089-f001:**
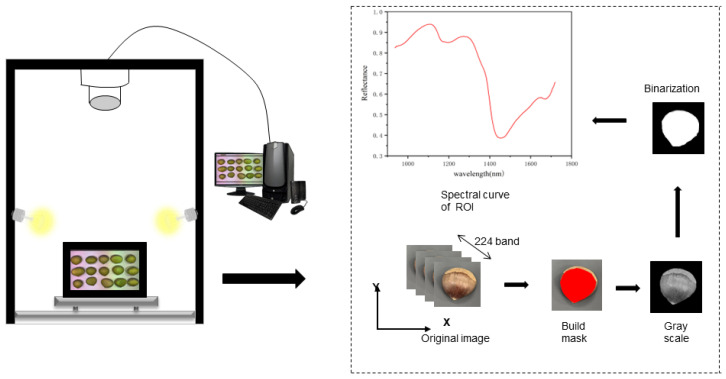
Schematic diagram of hyperspectral acquisition system.

**Figure 2 foods-12-02089-f002:**
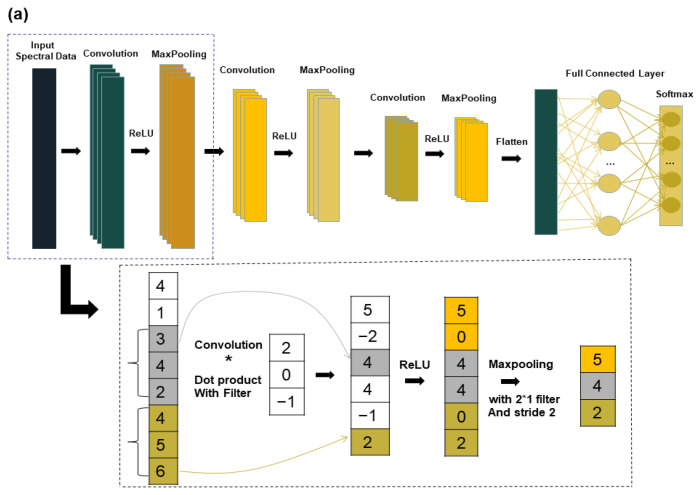
Deep learning model structure diagram: (**a**) CNN structure diagram; (**b**) LSTM structure diagram.

**Figure 3 foods-12-02089-f003:**
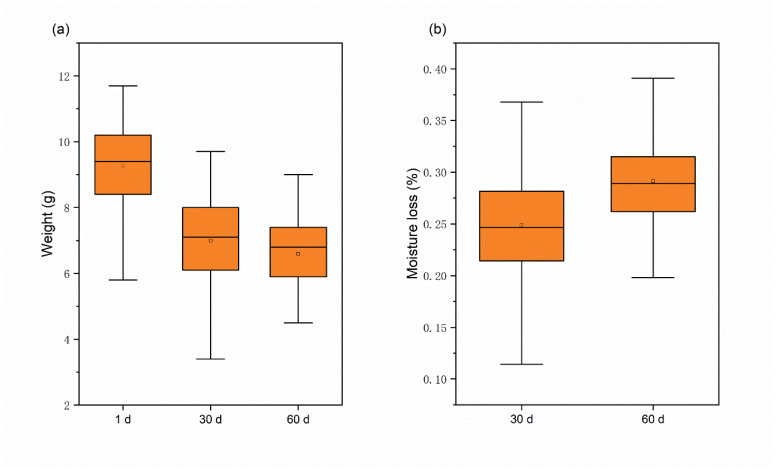
Weight and moisture loss of chestnuts: (**a**) weight variation; (**b**) moisture loss.

**Figure 4 foods-12-02089-f004:**
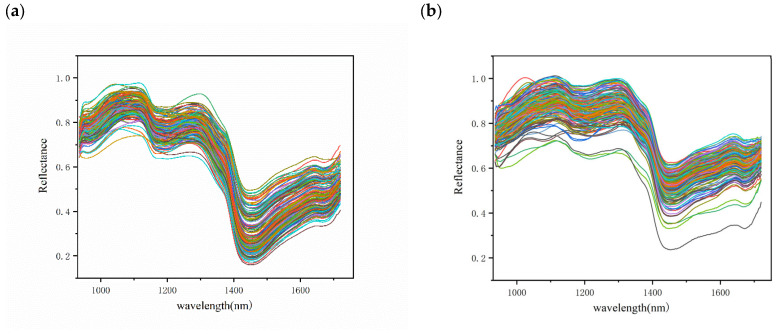
Reflectance spectra of different quality chestnuts: (**a**) fresh chestnuts; (**b**) sub-fresh chestnuts; (**c**) rotten chestnuts; (**d**) average spectrum.

**Figure 5 foods-12-02089-f005:**
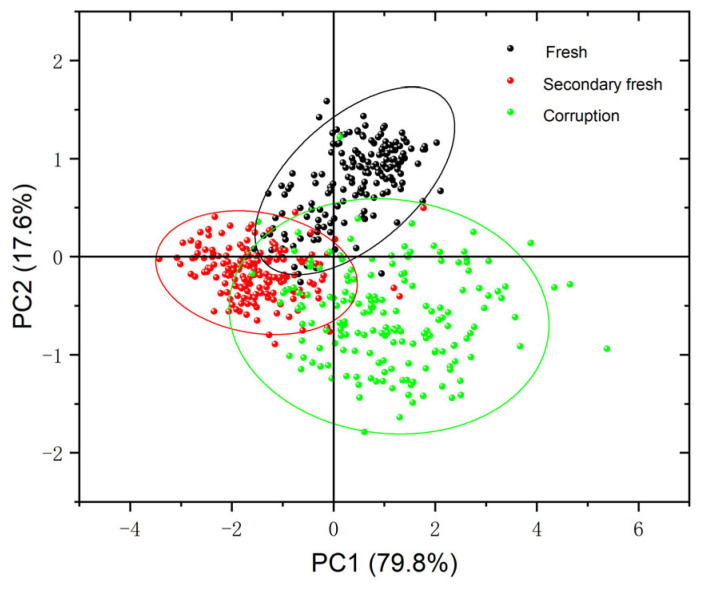
PCA score graph of chestnut samples.

**Figure 6 foods-12-02089-f006:**
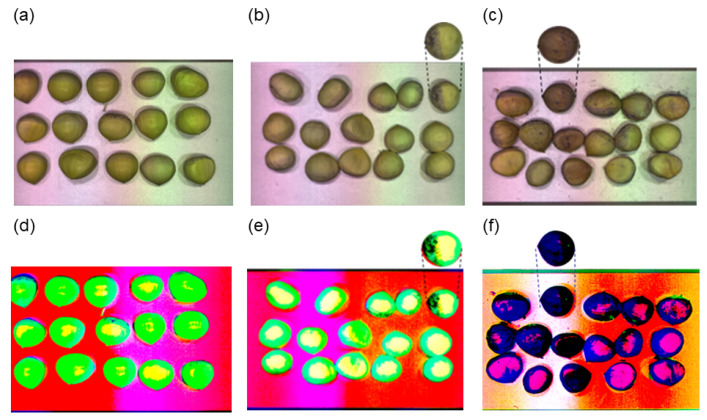
Sample raw images and principal component images: (**a**) fresh chestnuts; (**b**) sub-fresh chestnuts; (**c**) rotten chestnuts; (**d**–**f**) images of the first 3 principal components of the corresponding quality chestnut.

**Figure 10 foods-12-02089-f010:**
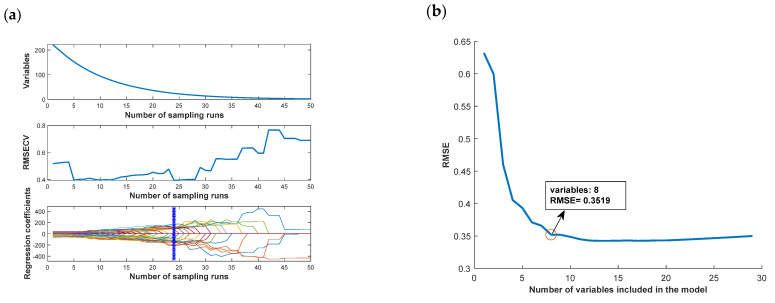
Important wavelength extraction results: (**a**) CARS algorithm; (**b**) SPA algorithm; (**c**) UVE algorithm; (**d**) important wavelength distribution.

**Figure 12 foods-12-02089-f012:**
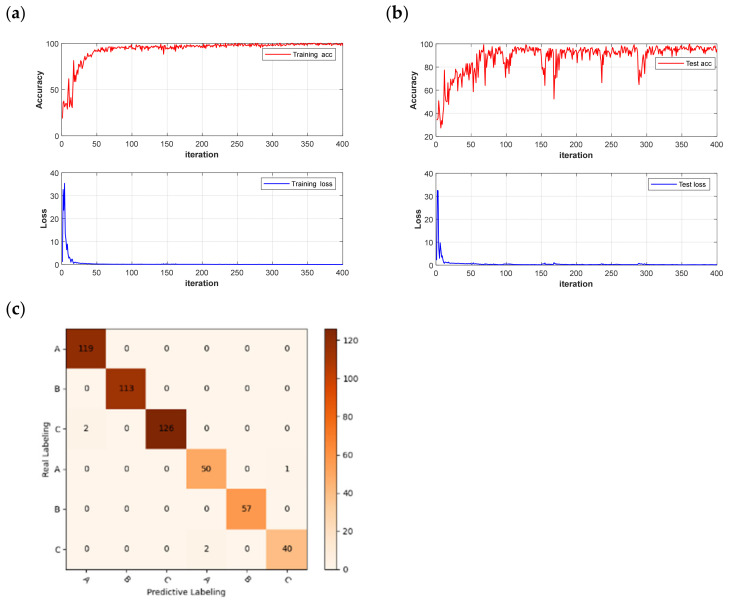
Model results and confusion matrix: (**a**) CNN training set accuracy and loss function; (**b**) CNN test set accuracy and loss function; (**c**) FD-UVE-CNN model confusion matrix.

**Table 1 foods-12-02089-t001:** Results of sample weight measurements.

Time	Number of Samples	Max	Min	Mean	SD ^1^
1 d	170	11.7	5.8	9.28	1.35
30 d	170	9.7	3.4	6.99	1.27
60 d	170	9	3.2	6.59	1.15

^1^ Standard Deviation.

## Data Availability

The datasets generated for this study are available on request to the corresponding author.

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
