# Peer review of "Feasibility Study of Combining Hyperspectral Imaging with Deep Learning for Chestnut-Quality Detection"

_foods, 2023, doi:10.3390/foods12102089_

Round 1
Reviewer 1 Report
The authors descibe methods using hyperspectral imaging to determine chestnut quality using PCA, image pre-processing, and deep learning methods. They convincingly demonstrate that using a first derivative - uninformative variable elimination - convolutional neural network algorithm they can accurately classify chestnut quality with 97.33% accuracy in 11 seconds processing time. I have the following observations the authors should address:
Line 28: should be HSI and not HIS. I realize Word does this without any input from you.
Line 51: sentence is incorrect and needs some editing.
Line 80: what is “6 d of spectral data?” If you mean days spelling it out is preferable, and it’s not a very long word.
Fig. 1: how was the ROI determined? Did you take an area of the chestnut and find the mean spectrum?
Line 111: please add degree type, I assume C but best to specify.
Line 124: if this is supposed to be 1 nanometer please express as 1 nm or spell it out.
Line 140: it’s 1st not 1th when expressing ordinal numbers.
For figure 4 and 7, where spectral curves are shown, are these of a single pixel, multiple pixels, all the pixels of each ROI?
Line 181: I think RESECV is supposed to be RMSECV. You also don’t define this abbreviation. I assume this is root mean square error – cross validation.
Line 221: do you mean Eq. 6? I don’t see an Eq. 7.
Fig 3 has the same caption as Fig 2. Fig 3 summarizes moisture loss and weight of the chestnuts.
Figure 6, d-f: how did you map the colors into the 1st three components? Are these pseudo colors or, for example, did you map pc1,pc2, and pc3 into red greed blue or something similar?
Fig 8 caption: looks like you typed the euro symbol instead of (e)
English is not bad but could use some grammar editing.
Author Response
Dear Reviewer:
We would like to thank you and the reviewers for your careful reading, helpful comments and constructive suggestions, which greatly improved our manuscript entitled "Feasibility study of combining hyperspectral imaging with deep learning for chestnut quality detection".
We have carefully considered all the reviewers' comments and have revised our manuscript accordingly. The revisions are marked in red text in the revised manuscript.
Please refer to the attachment for detailed responses to the reviewers.

Reviewer 2 Report
foods-2389420-peer-review-v1
The authors present an interesting study of using hyperspectral imaging coupled with deep learning to detect defects in chestnuts. The authors’ work is appreciated but some details need to be clear.
Introduction
The authors need to add images for examples of different molds and defects of chestnuts.
The authors need to add numbers to the studies in the literature to show the performance of the models deduced before.
Line 78: R2 and not R2. Correct in the whole manuscript.
Lines 78 and 83: Be consistent of displaying the R2, r, accuracy values: either decimal or %
What would be the advantage of HSI over CT-x-ray that has already been used? You listed the cost and operational skills, in addition to the analysis, this is still an issue with HIS systems.
Materials and Methods
Why LSTM is used ? It is a recurrent NN. Are you interested in monitoring the defects over time?
Line 227: Figure 2 and not Figure2. Correct similar typos in the whole manuscript.
Results
Some figures are blur, look at Figure 4.Figure 10 Be consistent in terms of font size, type, etc.
Discussion
The authors still need to put more effort in comparing and justifying their results with previous work for chestnut using HIS or other sensors and even similar nuts
The English language needs moderate revision.
Author Response
Dear Reviewers:
We would like to thank you and the reviewers for your careful reading, helpful comments and constructive suggestions, which greatly improved our manuscript entitled "Feasibility study of combining hyperspectral imaging with deep learning for chestnut quality detection".
We have carefully considered all the reviewers' comments and have revised our manuscript accordingly. The revisions are marked in red text in the revised manuscript.
Please refer to the attachment for detailed responses to the reviewers.
